# The Role of the Circadian Rhythm in Dyslipidaemia and Vascular Inflammation Leading to Atherosclerosis

**DOI:** 10.3390/ijms241814145

**Published:** 2023-09-15

**Authors:** Balazs Csoma, Andras Bikov

**Affiliations:** 1Wythenshawe Hospital, Manchester University NHS Foundation Trust, Manchester M23 9LT, UK; csoma.balazs@phd.semmelweis.hu; 2Department of Pulmonology, Semmelweis University, 1083 Budapest, Hungary; 3Division of Immunology, Immunity to Infection and Respiratory Medicine, School of Biological Sciences, Faculty of Biology, Medicine and Health, The University of Manchester, Manchester M13 9PL, UK

**Keywords:** circadian rhythm, circadian clocks, sleep disorders, sleep wake disorders, work schedule tolerance, immunity, inflammation, atherosclerosis, metabolic diseases, chronotherapy

## Abstract

Cardiovascular diseases (CVD) are among the leading causes of death worldwide. Many lines of evidence suggest that the disturbances in circadian rhythm are responsible for the development of CVDs; however, circadian misalignment is not yet a treatable trait in clinical practice. The circadian rhythm is controlled by the central clock located in the suprachiasmatic nucleus and clock genes (molecular clock) located in all cells. Dyslipidaemia and vascular inflammation are two hallmarks of atherosclerosis and numerous experimental studies conclude that they are under direct influence by both central and molecular clocks. This review will summarise the results of experimental studies on lipid metabolism, vascular inflammation and circadian rhythm, and translate them into the pathophysiology of atherosclerosis and cardiovascular disease. We discuss the effect of time-respected administration of medications in cardiovascular medicine. We review the evidence on the effect of bright light and melatonin on cardiovascular health, lipid metabolism and vascular inflammation. Finally, we suggest an agenda for future research and recommend on clinical practice.

## 1. Introduction

Cardiovascular diseases (CVD) are the leading causes of morbidity and mortality worldwide. Sleep disturbances, including abnormal sleep quantity and quality as well as diagnosed sleep disorders have all been associated with the development and worsening of CVD [1,2,3]. Most particularly, shortened sleep time was related to cardiovascular morbidity and mortality in large observational studies [1]. However, shortened sleep time could be attributed to insomnia (difficulties in initiating and/or maintaining sleep) and circadian rhythm disorders, particularly delayed phase sleep disorder (DPSD), work shifts, and voluntary factors (i.e., social jetlag). Physiologically, adults need 7–9 h of sleep. In Western cultures, the usual bedtime is between 10 pm and 12 am with corresponding wake up times at 6–8 am. Normally, people fall asleep within 30 min, but minimal variations may exist depending on socioeconomic conditions. In patients with DPSD, the habitual sleep-wake time is delayed, usually by more than two hours, relative to socially acceptable timing. Due to work and school commitments, the wake-up time is the same as normal, or only slightly delayed. This results in shortened sleep. When circumstances allow (on weekends or holidays), patients with DPSD tend to have normal sleep duration with delayed bedtimes and wake-up times (Figure 1). Similarly, shift workers have discrepancies between body clock homeostasis and social life.

Different preferences for the habitual time of wakefulness and sleep (called chronotypes) are driven by the circadian system. In broad terms, the circadian system is responsible for the adaptations to the diurnal variability of the homeostatic needs. It consists of a central clock in the suprachiasmatic nucleus (SCN) part of the brain and the peripheral clock genes in cells. The SCN receives projections from the retina and the environmental light synchronises the internal approximately 24 h rhythm with the exact 24 h rhythm of the environment. The SCN regulates the main mechanisms of metabolism, including food intake and the production of hormones and enzymes participating in glucose, protein and lipid metabolism (Figure 2). The SCN projects to various parts of the brain, including the ventrolateral preoptic nucleus, subparaventricular zone and dorsomedial hypothalamus (orchestrating sleep), pineal gland (responsible for melatonin production), pituitary gland (responsible for growth hormone production), arcuate nucleus (responsible for food intake), paraventricular nucleus (responsible for cortisol production) and the sympathetic, parasympathetic and dopaminergic systems [4]. It also regulates the activity of the peripheral tissue clocks via regulating body temperature, food intake and neuronal mechanisms. Various factors tune the rhythmic activity of the central clock which are called “Zeitgebers”. The most important of these is light; however, temperature, sleep, exercise and food intake can also modulate the central clock. Light suppresses melatonin, which is responsible for initiating sleep, but melatonin also regulates SCN. In clinical practice, both bright light and melatonin are used to influence the central clock. Bright light administered in the morning advances, and in the afternoon delays, sleep onset. In contrast, melatonin administered in the morning (when melatonin production declines) delays, and in the evening (when melatonin production increases) advances, sleep onset [5].

Peripheral clocks are located in all cells, including the bowel, liver, muscles, vascular and adipose tissues, affecting both lipid metabolism and inflammatory response (Figure 2). Whilst the central clock is predominantly affected by exposure to light and melatonin, peripheral clocks adapt to the local needs (i.e., diurnal variations in metabolite levels, temperature, hormones, etc). The core components of the peripheral molecular clock include the Brain and Muscle Aryl hydrocarbon receptor nuclear translocator-like 1 (BMAL1) and Circadian Locomoter Output Cycles Kaput Protein (CLOCK) transcription factors. Together they form a heterodimeric complex which affects the rhythmic expression of various clock-controlled genes, several of them participating in inflammation and metabolism. The heterodimer also activates the *Period* (*PER1*, *PER2* and *PER3*) and *Cryptochrome* (*CRY1* and *CRY2*) genes which inhibit the CLOCK-BMAL1 complex. The degradation of PER and CRY proteins releases the repression on *BMAL1* and *CLOCK*, leading to its cyclic activity. The CLOCK-BMAL1 complex also activates genes encoding the REV-ERBs nuclear hormone receptors (REV-ERBα and REV-ERBβ) and retinoic acid-related orphan receptors (RORα and RORβ). Whilst RORs serve as activators for CLOCK-BMAL1, REV-ERB is another repressor. Apart from regulating their own activation, these transcriptional activators and repressors control a significant number of clock-controlled genes driving the rhythmic expression of biological pathways. For the detailed posttranscriptional and posttranslational as well as metabolic regulation of the molecular clock, we refer to comprehensive reviews on the topic [6,7].

Dyslipidaemia and vascular inflammation are the two hallmarks of atherosclerosis and subsequent cardiovascular disease [8]. Various triggers such as dyslipidaemia, haemodynamic stress and decreased nitric oxide availability lead to endothelial cell activation and dysfunction. Activated endothelial cells produce cytokines and chemokines and express adhesion molecules that induce monocyte recruitment, activation and trafficking into the vascular bed. Here, activated macrophages will undergo proinflammatory M1 polarisation that is facilitated by the uptake of oxidized low-density lipoprotein (oxLDL) particles and, ultimately, macrophages turn into foam cells. Foam cells produce cytokines and chemokines to further enhance inflammatory cell influx, as well as growth factors which promote extracellular matrix formation. In addition, heightened vascular inflammation facilitates cell necrosis and limits reverse cholesterol transport. The resulting necrotic core, together with the increased extracellular matrix, form the atherosclerotic plaque that is an active inflammatory organ and serves as a surface for the coagulation cascade [8,9] (Figure 3).

Dyslipidaemia and vascular inflammation are affected by both central and peripheral clocks; therefore, understanding the impact of the circadian rhythm and the interrelation between the two systems could be of clinical impact. In this review, we will focus on the effect of circadian rhythm sleep-wake disorders on CVD and the molecular mechanisms leading to atherosclerosis development. We will review the evidence of how treatment of circadian misalignment could impact CVD. Of note, cardiac disease may also modulate circadian rhythm via denervating the pineal gland, resulting in reduced melatonin production [10]. However, the reciprocal mechanisms will not be discussed in the manuscript.

## 2. Circadian Rhythm Sleep-Wake Disorders and Cardiovascular Diseases

Blood pressure (BP), heart rate (HR), platelet aggregability, circulating catecholamine and cortisol levels, and vascular endothelial function show significant diurnal variability [11,12,13,14,15,16], with HR, adrenaline, cortisol levels and platelet activation reaching their highest levels in the early morning hours [17]. These observations provide a context and explanation for the phenomenon that CV events occur most frequently around the transition from the resting to the active phase [18]. According to large-scale epidemiological data, heart attack, cerebrovascular thrombotic events and sudden cardiac death are more prevalent around the time of waking up [19,20,21]. However, it is challenging to determine whether environmental and behavioural factors or an endogenous circadian rhythm contributes to a greater extent to this observation. These two tightly interrelated but different factors can be separated by highly controlled laboratory settings such as constant routine (CR) and forced desynchronisation (FD) [22,23,24]. In addition to epidemiological data and stringent laboratory protocols, mechanistic in vivo and animal models also provide insights into this complex relationship. In the following, we briefly summarise the available evidence linking circadian misalignment to cardiovascular diseases.

### 2.1. Normal Within-Day Variations in CV Functions

The diurnal variation in BP has a substantial clinical relevance [25]. A 24 h measurement of healthy adults shows that BP is the lowest during sleep and shows a marked rise after waking up [26]. The importance of this physiological pattern is highlighted by the adverse outcomes of patients with non-dipping hypertension [27,28], i.e., the loss of reduced BP during sleep, and by the more than 20% increased hazard of cerebrovascular events in hypertensive patients with exaggerated BP surge after awakening [29,30]. However, based on controlled CR and FD studies, the endogenous circadian rhythm of BP shows a zenith in the evening and its lowest in the morning [12,31]. Consequently, the morning surge might be due to behavioural factors such as changes in posture, psychological stress and physical exercise. This hypothesis is supported by the study of Scheer et al., in which the authors measured in an FD protocol a two-fold increase in catecholamine release in response to physical exercise in the morning hours compared to that at night [31].

Similarly, the hormones regulating BP, such as adrenaline, noradrenaline and cortisol, as well as the HR, also exhibit circadian variability [17,24]. For example, circulating levels of catecholamines and, consequently, the sympathetic tone peak around the middle of the day. In contrast, cortisol, of which the production is directly influenced by the central clock, reaches its highest point in the early morning hours [31], which presumably prevents the immune system from overactivation during the transition to activity, which is evolutionarily associated with a higher risk of injuries [32].

A similar evolutionary reason can explain the early morning peak in platelet aggregability, and in the levels of plasminogen activator inhibitor-1 [16,33,34,35,36]. Although these mechanisms contributed to the survival of our ancestors, nowadays these might increase the risk of thromboembolic events. Supporting this, aspirin, a potent platelet activation inhibitor, was proven by the Physicians’ Health Study to reduce the incidence of early morning myocardial infarction (MI) by 60%, whereas the reduction was only 34% in the remaining hours of the day [37].

Another important contributor to CV health is the vascular endothelium [38,39,40]. The vascular endothelium plays a crucial role in regulating blood pressure by modulating vascular resistance through the production of nitric oxide [41,42,43], and also has an important function in regulating blood clotting [44,45]. Vascular endothelial function is impaired in humans after waking up in the morning and, interestingly, even after waking from naps [46,47,48]. In addition, *PER2* clock gene-modified animal studies have demonstrated an association between circadian misalignment and vascular aging, decreased proliferation of the vascular endothelium and impaired vascular relaxation [49,50].

The cardiac function is regulated by the circadian rhythm via both central mechanisms and the molecular clock [17,51,52]. The central pathway involves the modulation of cardiomyocytes and vascular endothelial cells via sympathetic, parasympathetic and endocrine (i.e., corticosteroids) routes [51,52]. Approximately 13% of genes in the heart exhibit circadian variations in their expression [53]. The most studied gene is *BMAL1*, and its deletion completely abolishes circadian variability in cardiac function [54]. In addition, *BMAL1* deletion in mice leads to dilated cardiomyopathy, heart failure and early death [55,56], and affects the rhythmic expression of Na^+^ and K^+^ channels [57].

Furthermore, *BMAL1* deletion in vascular smooth muscle cells leads to the termination of within-day changes in pulse pressure [58]. Finally, modification of the *PER1* and *PER2* or *CRY1* and *CRY2* genes in kidney cells causes salt-sensitive hypertension, highlighting the importance of circadian rhythms in aldosterone synthesis and release [59,60].

### 2.2. Clinical Consequences of Circadian Disorders on Cardiovascular Health

Normal circadian rhythms are commonly disturbed by either social and behavioural habits like shift work or jetlag, or by sleep-wake disorders such as DPSD [17]. A persistent mismatch between the internal circadian rhythm and external environmental and behavioural cycles, known as circadian misalignment, has been linked to an increased risk of metabolic syndrome, cardiovascular diseases and cancer [61,62,63,64,65,66]. The exact link between circadian misalignment and cardiovascular diseases has been studied in preclinical mechanistic studies in humans. For example, Morris et al. have demonstrated in a shift working laboratory model of 12 h with inverted behavioural and environmental cycles thatshort-term circadian misalignment increases 24 h BP levels, reduces the usual night-time decrease in BP and raises systemic inflammatory markers such as interleukin-6 (IL-6), C-reactive protein (CRP) and tumour necrosis factor-α (TNF-α) in healthy adults [62]. Another study also revealed a significant increase in cardiovascular risk in shift workers owing to circadian misalignment [63]. Moreover, in a study using FD protocol, circadian misalignment reduced leptin, increased glucose and insulin levels, reversed the normal daily cortisol rhythm and also increased BP [23]. A recent meta-analysis on 39 observational studies concluded that evening chronotype was associated with higher blood glucose, low-density lipoprotein (LDL) cholesterol and triglyceride values, explaining the increased cardiovascular risk in these people [67].

Additionally, individuals experiencing circadian misalignment may also suffer from sleep disruption, further elevating the cardiovascular risk [68,69,70]. Sleep deprivation augments proinflammatory responses, elevates heart rate, impairs glucose tolerance and promotes vascular dysfunction [69,71,72]. Research has demonstrated that combining circadian misalignment with sleep deprivation in experimental studies leads to heightened sympathetic activation and elevated heart rates, and results in a higher level of inflammatory markers than those seen in cases of circadian misalignment and sleep deprivation alone [73,74].

Moreover, even moderate disruption compared to working at night can have a noticeable impact on cardiovascular risk. For example, a systematic review of epidemiologic data shows that a one hour difference in sleep schedule during the transition to daylight saving time in the spring can elevate the occurrence of MI up to 29% [75]. Furthermore, mild but persistent disruptions to sleep, such as variability in sleep duration and timing between weekdays and weekends (social jetlag), high day-to-day variations or exposure to light during sleep, have been linked to an increased risk of cardiometabolic diseases, and this risk tends to accumulate over time [76,77,78].

In addition to increasing the cardiovascular risk, circadian misalignment can also hinder the recovery from previously occurring cardiovascular events. Alibhai et al. demonstrated that short-term disturbance to the circadian rhythm following an MI negatively affects the long-term cardiac function in mice [79].

In conclusion, disruptions in normal circadian rhythms have been associated with increased risks of cardiovascular diseases.

## 3. The Role of Circadian Rhythm and Dyslipidaemia

Blood lipid values are determined by food intake, degradation in the intestine by bile acids and lipases, absorption by enterocytes, degradation in the blood vessels by lipoprotein lipases as well as adipose tissues by adipose triglyceride lipase and hormone sensitive lipases, de novo synthesis in the liver and removal by the peripheral cells and the liver [8]. It is important to note that lipids are carried by albumin and apolipoproteins in the blood and lymphatic system. Lipids and apolipoproteins combine to form lipoproteins, including chylomicrons, very low-density lipoprotein (VLDL), LDL, intermediate-density lipoprotein (IDL) and high-density lipoprotein (HDL). The carrier proteins are produced in the liver and intestines.

Lipids, albumin and apolipoproteins all show diurnal variation in animal models [80,81] and humans [82]. In addition, an experimental night shift also leads to increased serum triglycerides in healthy young adults [83]. In a comprehensive review of human data, Poggiogalle et al. concluded that although most studies reported significant circadian variation in serum lipid levels, the results were inconsistent on the magnitude of variations and the time of peaks and nadirs. Apart from lifestyle factors, such as diet and exercise, medications and comorbidities, significant inter-sex differences were noticed [82]. These conclusions highlight an important clinical aspect, namely, when considering personalising anti-dyslipidaemia treatment based on chronotype, individual circadian lipid variations need to be measured at baseline to account for within-day variations.

The central clock has a significant effect on hunger, appetite and food intake [84]. The SCN projects signals directly to the arcuate, ventromedial and dorsomedial hypothalamic, paraventricular and lateral hypothalamic nuclei in the brain, which are the main central regulators of energy homeostasis [85,86]. As a consequence, the levels of leptin [87], ghrelin [88], glucagon-like peptide 1 [89], peptide YY [90] and cholecystokinin [91], the main hormones regulating appetite, show diurnal rhythms with highest leptin (anti-appetite) and lowest ghrelin (pro-appetite) levels at night.

The intestinal absorption of lipids depends on bowel motility, degradation by bile acids and enteral lipases and transporters expressed on enterocytes. Gastrointestinal motility [92] and the production of bile acids and pancreatic lipase show circadian patterns [93]. For example, farnesoid X Receptor, which is a main regulator of bile acid synthesis, is directly regulated by BMAL1 [94]. In addition, REV-ERB encoding gene knock out mice showed reduced Cyp7a1 expression, which is an essential enzyme for bile acid production [95]. In line with this, experimental circadian rhythm disruption led to an altered expression of enzymes and transporters participating in bile acid secretion [96].

CD36 and fatty acid transport proteins participate in free fatty acid (FFA) and monoacylglycerol absorption, whilst CD36 and Niemann–Pick C1-like protein (NPC1LP) are involved in free cholesterol uptake [8,97]. In addition, ATP-binding cassette-binding proteins G5 (ABCG5) and G8 (ABCG6) are expressed in the apical membranes of the enterocytes and are responsible for free cholesterol efflux towards the intestine [97]. Mice expressing the *CLOCK*∆19/∆19 mutant protein have higher levels of CD36 and NPC1LP [98]. In addition, *BMAL1* knock out mice showed increased NPC1LP and decreased ABCG5 and 8 expressions [99]. These studies suggest significant circadian influence for enteral lipid uptake and release.

Following absorption, lipids form chylomicrons in the endoplasmic reticulum of the enterocytes by binding to the apoB-48 apolipoprotein [97]. This formation is mediated by the microsomal triglyceride transfer protein (MTP). *CLOCK* knock out mice showed increased MTP expression, suggesting direct circadian control on MTP [100]. Chylomicrons undergo further modification by taking apolipoprotein A-IV, which is also directly influenced by BMAL1 [101] and then released into the lymphatic system.

Free fatty acids are liberated from chylomicrons by the lipoprotein lipase. An increased expression of lipoprotein lipase was shown in *REV-ERBα* knock out mice, and CLOCK further upregulated its expression suggesting direct control by the *CLOCK* gene [102]. Lipoprotein lipase activity could also be indirectly regulated by the molecular clock. The proliferator-activated receptor γ (PPARγ), an important activator of the lipoprotein lipase [103], was reduced in BMAL1 deficient cells [104]. In line with this, a circadian variation in lipoprotein lipase was reported [105]. The remnant chylomicrons are picked up by the liver by LDL receptors (LDLR). Ma et al. reported that the deletion of *BMAL1* led to a decreased LDLR expression [106].

Many lines of evidence suggest the circadian control of hepatic lipid synthesis. Sterol regulatory element binding proteins (SREBPs) regulate cholesterol, FFA, triglyceride and phospholipid synthesis in the liver [107]. The expression of SREBPs were reduced in *REV-ERB* knock out mice [95] and *BMAL1* knock out cells [104], whilst *CLOCK* knock out affected their rhythmic production [108]. β-Hydroxy β-methylglutaryl-CoA (HMG-CoA) reductase, an important enzyme in cholesterol synthesis and the main target for statin therapy, shows diurnal rhythmicity with the highest expression at midnight [109,110]. The enzymes participating in cholesterol synthesis are regulated through PPARs [111] which are regulated by BMAL1 [104]. Cholesterol is released through bile acids, which, as mentioned above, are under the control of the molecular clock [96].

Enzymes participating in hepatic triglyceride synthesis also show circadian control, as demonstrated in *PER* knock out mice [112]. BMAL1 deficient mice had higher VLDL production and increased hepatic VLDL concentrations [99]. MTP plays a central role in VLDL formation as it binds apoB-100 to VLDL [8]. Both BMAL1 [99] and CLOCK [98,100] deficient mice showed higher MTP expression. Increased VLDL concentrations could also be due to increased apolipoprotein A-IV production in the liver in a BMAL1-deficient mice model [101]. Circulating VLDL particles are degraded to IDL and LDL via lipoprotein and hepatic lipases, which were reported to show circadian rhythm [105].

Free fatty acids and glycerol are also liberated from adipose tissues. On the one hand, adipogenesis was linked to *PER3* gene deletion [113]. On the other hand, Shostak et al. demonstrated that the expression of multiple enzymes participating in FFA and glycerol release and triglyceride synthesis is altered in *CLOCK*∆19 and *BMAL1*^−/−^ mutant mice. Most importantly, both adipose triglyceride lipase and hormone sensitive lipases were directly affected by the *CLOCK* and *BMAL1* genes [114]. The hormone sensitive lipase is blocked by insulin, which also shows strong circadian influence [115]. The increased liberation of FFA from adipose tissue in BMAL1 deficient mice leads to ectopic fat tissue formation in the liver and skeletal muscle [116].

The LDL particle is the main carrier of cholesterol. If its uptake is delayed by the peripheral cells or liver, it goes through modifications, such as oxidisation by reactive oxygen species, leading to atherogenic oxLDL particles [8]. The hepatic uptake of LDL is driven by hepatic LDLR which shows circadian influence [106]. Circulating LDL particles diffuse through the endothelial barrier and are taken up by vascular macrophages via scavenger receptors. Mice expressing the *CLOCK*∆19/∆19 mutant protein showed increased CD36 and scavenger receptor A1 expression, contributing to the development of atherosclerosis [98]. ATP-binding cassette A1 (ABCA1) is an important constituent of HDL and is responsible for the reverse cholesterol transport [8]. Mice expressing *CLOCK*∆19/∆19 mutant protein showed a reduced ABCA1 expression [98].

SCN affects cortisol production via projecting afferents to the paraventricular nucleus [117]. High cortisol levels lead to insulin resistance, liberate free fatty acids from adipose tissue, increase VLDL production and decrease LDL uptake by the liver [118]. Projections from the SCN also affect melatonin production [4] which stimulates insulin secretion [119]. Insulin stimulates lipoprotein lipase, contributing to free fatty acid influx towards adipose tissue and blocks lipolysis there [8]. Finally, SCN projects afferents to hypophysis, affecting growth hormone production [120]. Growth hormone facilitates lipolysis in the adipose tissue [121]. Of note, growth hormone production is also directly affected by sleep as it is mainly produced during deep (N3) sleep [122].

## 4. The Role of Circadian Rhythm and Vascular Inflammation

Vascular inflammation is commonly observed in patients with circadian misalignment as it is linked to an increased risk of dyslipidaemia, endothelial dysfunction and, as will be outlined below, immune dysregulation [123]. Furthermore, vascular inflammation, in turn, contributes to all the aforementioned factors and, importantly, promotes atherosclerosis (Figure 3) [124].

### 4.1. Circadian Control of Immune Functions

Similarly to lipid metabolism, the immune system is also regulated by the central and molecular clocks. The former is driven predominantly through the endocrine system [32,51,71]. The molecular clock has an impact on immune functions and plays a role inflammation. For example, the BMAL1-CLOCK heterodimer represses the expression of several chemokines necessary for leukocyte trafficking and activation, including Ccl2, Ccl8 and S100a8, thereby generating time of day variations in leukocyte abundance and migration [125]. Additionally, BMAL1 plays an important role in the circadian regulation of other inflammatory mediators such as IL-1β, IL-6 and TNF-α [54,126]. Moreover, it has also been shown that the lack of BMAL1 leads to the proinflammatory activation of macrophages with an elevated production of reactive oxygen species, hypoxia-induced factor-1α, IL-1β and IL-6 [126]. Furthermore, *BMAL1* knock out mice have increased microRNA miR-155 production and elevated nuclear factor-κB (NFκB) activity [127]. Overall, BMAL1 appears to play a primarily anti-inflammatory role, although it is important to note that some of the effects are not directly related to circadian rhythms and are instead results of downstream effects.

In contrast, CLOCK was found to have proinflammatory properties, interacting with the p65 subunit of NF-κB and upregulating its transcriptional activity [128]. Meanwhile, PER1 and PER2 clock proteins seem to have contradictory effects, with PER1 as mostly anti-inflammatory and PER2 augmenting inflammation. In genetic knock out murine models, while *PER1* mutant mice produced higher levels of inflammatory mediators such as TNF-α, IL-1β, IL-6 and Ccl2, in *PER2* knock out specimens, the expression of interferon-γ (IFN-γ) and IL-1β was reduced [129,130]. Moreover, CRY proteins have anti-inflammatory functions. In *CRY1* and *CRY2* knock out animals, an increased production of TNF-α, IL-6, IL-1β, Cxcl-1 and inducible nitric oxide synthase enzyme was reported [131,132]. Finally, REV-ERBs and RORs, which also regulate BMAL1 transcription, act mainly against inflammatory processes. REV-ERBα suppresses the production of IL-6 in macrophages and represses the NLRP3 inflammasome [133,134,135]. Importantly, REV-ERBα also hinders Ccl2 production, which plays a crucial role in macrophage mobilisation and activation [136]. RORα also controls the production of TNF-α and IL-6 in macrophages and mast cells and increases the expression of an inhibitor of the NF-κB [137,138].

### 4.2. Circadian Misalignment and Vascular Inflammation

Vascular inflammation is a hallmark of atherosclerosis [8,123,124]. As outlined above, circadian rhythm plays a pivotal role in the regulation of macrophage migration, activation and proliferation, and many of the molecular mechanisms discussed are involved in the pathogenesis of atherosclerosis. Indeed, Ccl2 production, which exhibits a rhythmic pattern, is paramount in the pathogenesis of atherosclerosis and early plaque generation [139,140]. Ccl2 is necessary for the recruitment of Ly-6C^high^ monocytes and macrophages and its expression is regulated by BMAL1 and REV-ERBα [141]. It has been shown that the deletion of *BMAL1* in myeloid subsets in *ApoE* knock out mice accelerates atherogenic processes through increasing the recruitment of monocytes and M1 macrophage polarisation [142]. After increased migration from the bone marrow, macrophages in the blood require adhesion molecules to be present on vascular endothelial cells to extravasate [8]. The expression of certain adhesion molecules, such as the intercellular adhesion molecule-1 (ICAM-1) and vascular cell adhesion molecule-1 (VCAM-1), is controlled by CLOCK and CRY proteins [143]. Furthermore, proinflammatory cytokine production by activated macrophages (IL-1β, IL-6, IL-12, and TNF-α) also exhibits circadian regulation [144]. The phagocytic capacity of macrophages and the ability to transport lipoproteins and cholesterol to ApoA1 are mediated by CLOCK, as demonstrated in *CLOCK* and *ApoE* knock out mice whose uptake was increased, while the efflux was also impaired [98]. Moreover, the synthesis of nitric oxide, an important molecule with vasodilatory and immunoregulatory capabilities, shows circadian rhythmicity, which is abolished in *BMAL1*^−/−^ mice. Simultaneously, in *BMAL* knock out mice, the production of superoxide anions, a potent reactive oxygen species, increases, further aggravating atherosclerosis [145,146]. The relevance of these processes in atherogenesis is emphasised by the finding that the overexpression of *CRY1* and activation of REV-ERBα can suppress atherosclerotic development and decrease the levels of inflammatory mediators in *ApoE*^−/−^ mice, with much of these effects mediated through the repression of the NF-κB pathway [147,148].

The mechanistic models described above are also supported by epidemiological data. For example, secondary analyses of the Multi-Ethnic Study of Atherosclerosis, which is a large-scale prospective clinical study, have proven that circadian disruptions caused by sleep onset and timing irregularity are associated with a higher incidence of major cardiovascular adverse events [77]. Furthermore, even participants free of CV diseases during follow-up were more likely to have subclinical atherosclerosis if they experienced significant sleep irregularities [149].

In summary, vascular inflammation contributes to atherosclerosis through mechanisms involving the circadian control of immune functions. This has been supported by epidemiological evidence, which shows a correlation between circadian disruptions and cardiovascular risk, even in individuals without pre-existing cardiovascular disease, and experimental studies demonstrating the impact of circadian misalignment on vascular health.

## 5. Therapeutic Considerations

As our understanding of the complex relationship between circadian misalignment, inflammation, metabolic diseases and cardiovascular diseases has improved, the need to explore therapeutic interventions that influence and potentially restore circadian harmony has emerged. In the following, we will discuss chronotherapy, melatonin and bright light treatment, and some other emerging therapeutic options.

### 5.1. Chronotherapy

Chronotherapy may refer to two different approaches. Firstly, it may mean optimal timing of the intervention (i.e., taking medicine based on drug metabolism or peak of effectiveness). Secondly, in sleep medicine, it is a form of intervention for circadian rhythm disorders, involving the introduction of zeitgebers (i.e., light, melatonin, exercise, diet) and behavioural interventions (i.e., shifting bedtime while restricting wake-up time) to move the sleep onset based on patient’s needs.

Optimising feeding depending on the circadian rhythm may lead to weight loss, as it was shown in mice [150,151,152]. Importantly, timed diet affects both peripheral and central clocks [153]. The latter is particularly important as it coordinates homeostasis and may lead to metabolic misalignment and development of metabolic syndrome [153]. Apart from weight gain in general, time-restricted feeding influences circulating lipid levels [152] and lipid accumulation in the liver [154] in animals. Animal models have lately been confirmed in humans. Late eating was associated with increased hunger, ghrelin/leptin ratio, reduced waketime energy expenditure and lipolysis. Importantly, it also influenced the expression of inflammatory genes [155]. Interestingly, a late versus early meal seems to be more detrimental for triglyceride than cholesterol values [156]. The metabolic effect of a time-restricted diet has recently been summarised in topical and systematic reviews [157,158].

A few studies were conducted to compare if a morning or evening schedule of a particular administration of drugs is more effective [159]. Regarding cardiovascular disease, evening administration of ramipril [160], olmesartan [161], telmisartan [162] and nifedipine [163] was more effective than the morning dosage in reducing blood pressure, especially the nighttime values. Similarly, evening administration of simvastatin was more effective in reducing total cholesterol levels than taking it in the morning [164]; however, the results were not replicated in a smaller follow-up study [165]. Likewise, when simvastatin was combined with ezetimibe, morning administration was noninferior to the evening one [166]. As mentioned above, HMG-CoA reductase has its peak expression at night [109]. Additionally, simvastatin is metabolised by CYP3A4, which is significantly influenced by clock genes and, hence, its expression shows diurnal variability [167]. A systematic review on evening vs. morning administration of statins concluded that this effect is likely to be relevant only for short half-life drugs. The authors also emphasised that the administration of statins should follow patients’ preference due the potential impact of medication adherence [168]. Administering aspirin at bedtime significantly reduced the risk of preeclampsia in high-risk pregnant women compared to the morning dosage [169]. Whilst compared to the morning, bedtime aspirin was not superior in lowering blood pressure, but reduced platelet reactivity [170]. Furthermore, an evening application of RS102895, which is a CCR2 chemokine receptor antagonist, was more effective in ameliorating atherosclerosis than the morning administration [171].

### 5.2. The Effect of Melatonin

Numerous studies reported reduced melatonin levels in patients with coronary artery disease and chronic heart failure [172]. However, the effect of melatonin administration is inconclusive, which is due to its diverse and sometimes antagonistic effects on blood pressure, metabolism and inflammation.

Melatonin has various effects on the cardiovascular system through melatonin receptors (MT1 and MT2) that are located on cardiomyocytes and coronary arteries [172]. Pioneering studies showed that the administration of melatonin acutely decreased blood pressure [173,174,175], increased vagal tone and decreased sympathetic activity [176] in healthy humans, and has a vasodilatory function in rats [177]. In addition, melatonin taken at 4 pm advanced peak heart rate and heart rate variability compared to a placebo [178]. The chronic administration of melatonin in healthy, normotensive subjects resulted in blood pressure drops [179]. The effect of melatonin in hypertensive subjects is contrasting; some studies reported decreased BP values [180,181], others showing no effect [182,183], whilst one study concluded an increase in blood pressure and heart rate values [184]. In patients with non-dipper hypertension, a significant reduction in nighttime BP values were reported [185]. This was confirmed in another study; however, they also reported a corresponding increase in morning blood pressure values [186]. Focusing on high-risk populations, melatonin supplementation was associated with decreased BP in elderly [187] patients with metabolic syndrome [188], type 1 diabetes [189], type 2 diabetes [190,191] and non-alcoholic fatty liver disease (NAFLD) [192]. The contrasting results could be attributed to the antagonistic function of different melatonin receptors. Whilst melatonin exerts vasoconstriction on MT1, it relaxes blood vessels through MT2 [193].

The effect of melatonin on dyslipidaemia is also contradictory. In rats, melatonin reduced LDL-C and increased HDL-C levels [194]. Taken at night, it significantly improved postprandial triglyceride levels in healthy young males [195]. It did not change lipid results in patients with diabetes [196], but, notably, the glycaemic control was better after 5 months of treatment [196]. In another study in patients with type 2 diabetes, melatonin significantly increased HDL-C [191]. Furthermore, in patients with metabolic syndrome, it significantly reduced LDL-C [188]. In contrast, melatonin significantly increased triglyceride values in normolipidemic women [197]. Lastly, melatonin reduced the body mass index (BMI) of shift workers [198] and patients with diabetes [190] and NAFLD [192].

The results on preventing atherosclerosis are more coherent. Melatonin has been shown in vitro to reduce cholesterol synthesis and LDLR expression in monocytes [199]. Moreover, it also acts as a free radical scavenger [200] and stimulates antioxidant enzymes [201]; therefore, it is not surprising that melatonin prevented LDL oxidation [202] and oxLDL induced endothelial dysfunction in vitro [203].

Melatonin has diverse effects on the immune system. These include involvement in immune cell proliferation, survival, antigen presentation and the production of antibodies, cytokines and inflammatory molecules [204]. However, similarly to blood pressure and lipid metabolism, the immunological effects are contradictory. For example, in some studies it increased the anti-inflammatory IL-10 [205] and decreased the proinflammatory TNF-α [205,206] and IFN-γ [206] levels, whilst other studies showed an increased expression of proinflammatory TNF-α [207], IFN-γ [207,208], IL-1β [207], IL-2 [208] and IL-6 [208] cytokines following melatonin administration. A further study showed that it prevented cyclooxygenase-2 and inducible nitric oxide synthase activation in macrophages, resulting in decreased prostaglandin E2 and NO production [209]. A review on this topic concluded that the effects largely depend on immune system activation [204]. In line with this, the external administration of melatonin prevented smoking-induced inflammation in vitro [210]. Apart from the immune system, melatonin also acts on platelets by inhibiting their aggregation through suppressing thromboxane B2 release [211].

When administering melatonin to patients with NAFLD [192] and type 2 diabetes [191], a significant reduction in CRP was reported. The latter study also found significant improvement in antioxidant enzymes and a reduction in markers of oxidative stress [191]. Whilst melatonin did not change CRP values in patients with metabolic syndrome, a significant improvement in antioxidant capacity was noted [188].

### 5.3. The Effect of Bright Light

It is widely recognised that exposure to bright light in the morning advances, whilst in the evening delays sleep onset. Although it increases nighttime melatonin if applied in the morning [82], bright light is known to supress melatonin production in the evening [175]. This effect is spectral-dependent, and blue-enhanced light seems to be more potent than blue-depleted light [212].

Bright light in the evening was associated with higher systolic blood pressure values [175]. At night (00:00 to 04:00), it significantly increased heart rate, whilst no such effect has been noticed for bright light at daytime [213]. These results were replicated by another study [214]. In addition, evening bright light decreased heart rate variability at night [215]. Most notably, the effect of bright light on heart rate largely depends on light intensity [214] and wavelengths (blue light being more potent) [216].

The effect of light therapy on weight and metabolism is contradictory. It seems that if applied in the morning, it may reduce fat mass [217,218] and appetite [218] and increase adiponectin levels [217]; however other studies did not find such effect [219,220]. Whilst not influencing fasting triglyceride levels, morning bright light increased postprandial concentrations in patients with type 2 diabetes [221]. It is more likely that evening bright light is detrimental, as it was associated with increased BMI, triglyceride and LDL-C levels [222]. It seems that light therapy is effective on metabolic outcomes in a selected group of patients. For instance, a beneficial effect on weight loss was reported in patients with seasonal affective disorder [223]. The complex effect of light on metabolism, including the effect of intensity, wavelength and the time of exposure, has recently been summarised by Ishihara et al. [224].

The results on the effect of light therapy on inflammation are scarce. In a preliminary study, Elliott et al. reported a significant decrease in IL-6 and TNF-α levels following light therapy in the morning in patients with traumatic brain injury [225]. In contrast, no effect on IL-6 levels was observed following light therapy in patients with seasonal affective disorder [226].

### 5.4. Experimental Therapies

Nobiletin is an RORα agonist. It enhances the clock gene function, and treatment with nobiletin led to reduced weight gain and better glycaemic and lipidemic control; however, the effect was lost in *CLOCK*∆19/∆19 mice [227]. Promisingly, this compound attenuated VLDL production and corresponding atherosclerosis in LDLR-deficient mice [228]. SR8278 is an REV-ERBα selective antagonist. In an experimental model on isolated mouse hearts, the application of SR8278 prevented myocardial injury [229].

## 6. Conclusions and Suggestions for Further Research and Clinical Practice

Molecular mechanisms of lipid metabolism and inflammation are tightly regulated by the central and peripheral molecular clocks. Not surprisingly, cardiovascular physiology, lipid values and inflammatory molecules show a circadian rhythm. Circadian misalignment can therefore lead to an increased risk of cardiovascular disease (Figure 4). Although molecular research on cardiovascular and metabolic health is broad and robust, population-level data need to be more convincing to influence policy makers.

For instance, whilst evening chronotype, which is often related to shortened sleep, was associated with increased cardiovascular risk [230], it is important to note that abnormal sleep patterns may be caused by other common conditions, such as obstructive sleep apnoea or periodic limb movement disease [2,3]. These conditions can independently lead to cardiovascular disease via unique mechanisms. We therefore suggest validating population-level findings with objective sleep tests to delineate if cardiovascular risk is due to circadian misalignment or impaired sleep.

Secondly, it seems that bright light in the evening is detrimental for CVD, whereas in the morning, it might be beneficial. However, most studies did not investigate whether these generalised effects apply to certain population groups, such as those with circadian rhythm disorders and those with heightened cardiovascular risk.

Thirdly, melatonin appears to be beneficial to cardiovascular health. However, melatonin shows significant circadian rhythm, and certainly the efficacy likely depends on chronotype (peak/nadir of melatonin production). A randomised controlled trial designed to apply actigraphy or diurnal melatonin measurements before melatonin administration to personalise treatment is an unmet need.

Addressing lifestyle, comorbidities, usual bedtime, sleep time and wake up time are routine part of interviews during the assessment of sleep disorders. Acknowledging that circadian misalignment is an additional risk factor, clinicians should consider investigating (actigraphy) and treating (light therapy, chronotherapy, behavioural advice and melatonin) them in patients with high suspicion, even if they are diagnosed with another sleep disorder.

## Figures and Tables

**Figure 1 ijms-24-14145-f001:**
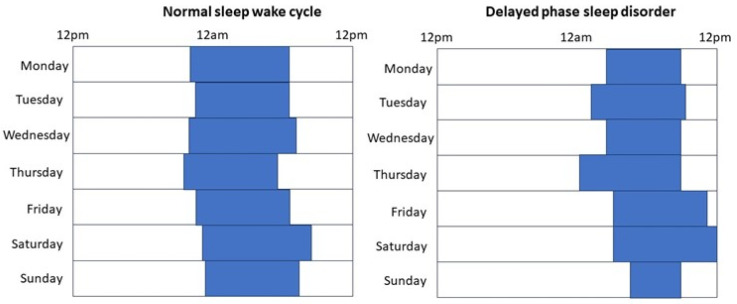
Schematic figures of subjects with normal sleep-wake cycle and with delayed phase sleep disorder.

**Figure 2 ijms-24-14145-f002:**
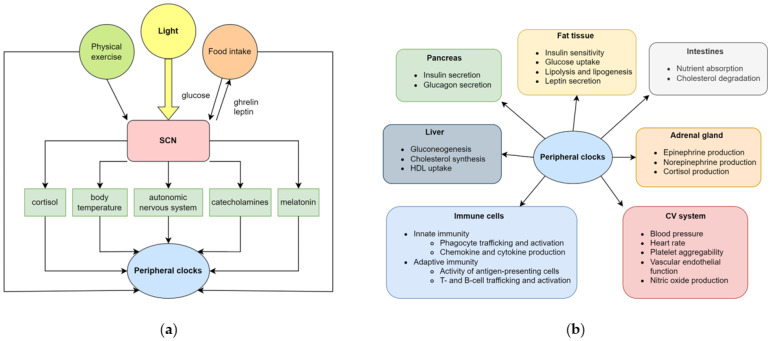
Central and peripheral clocks. (**a**) The central clock, which is located in the suprachiasmatic nucleus, entrains environmental and endogenous cues. The most important signal is the light, which is transmitted to the SCN through the retina and the retinohypothalamic tract. The SCN then synchronises the peripheral clocks, found in all cells, via multiple direct and indirect routes; (**b**)The peripheral clocks, regulated by the central clocks, control the circadian rhythm of several systems, including the endocrine, gastrointestinal, cardiovascular and immune system. Abbreviations: CV cardiovascular, HDL high-density lipoprotein, SCN suprachiasmatic nucleus. For detailed explanation, please see [4] and references provided in the text.

**Figure 3 ijms-24-14145-f003:**
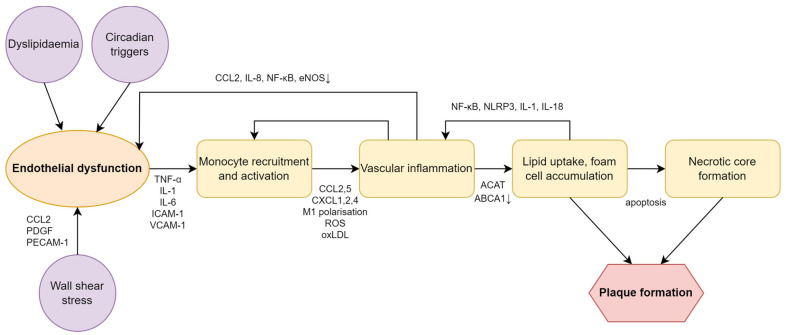
Schematic diagram of the pathogenesis of atherosclerosis. Downward arrows indicate decrease in level. Abbreviations: ABCA1 ATP binding cassette A1, ACAT acyl CoA:cholesterol acyltransferase, CCL C-C motif chemokine ligand, CXCL CXC chemokine ligand, eNOS endothelial nitric oxide synthase, ICAM-1 intercellular adhesion molecule-1, IL interleukin, NF-κB nuclear factor kappa-B, NLRP3 nod-like receptor family pyrin domain containing 3, oxLDL oxidised low-density lipoprotein, PDGF platelet-derived growth factor, PECAM-1 platelet endothelial cell adhesion molecule-1, ROS reactive oxygen species, TNF-α tumour necrosis factor-α, VCAM-1 vascular cell adhesion molecule.

**Figure 4 ijms-24-14145-f004:**
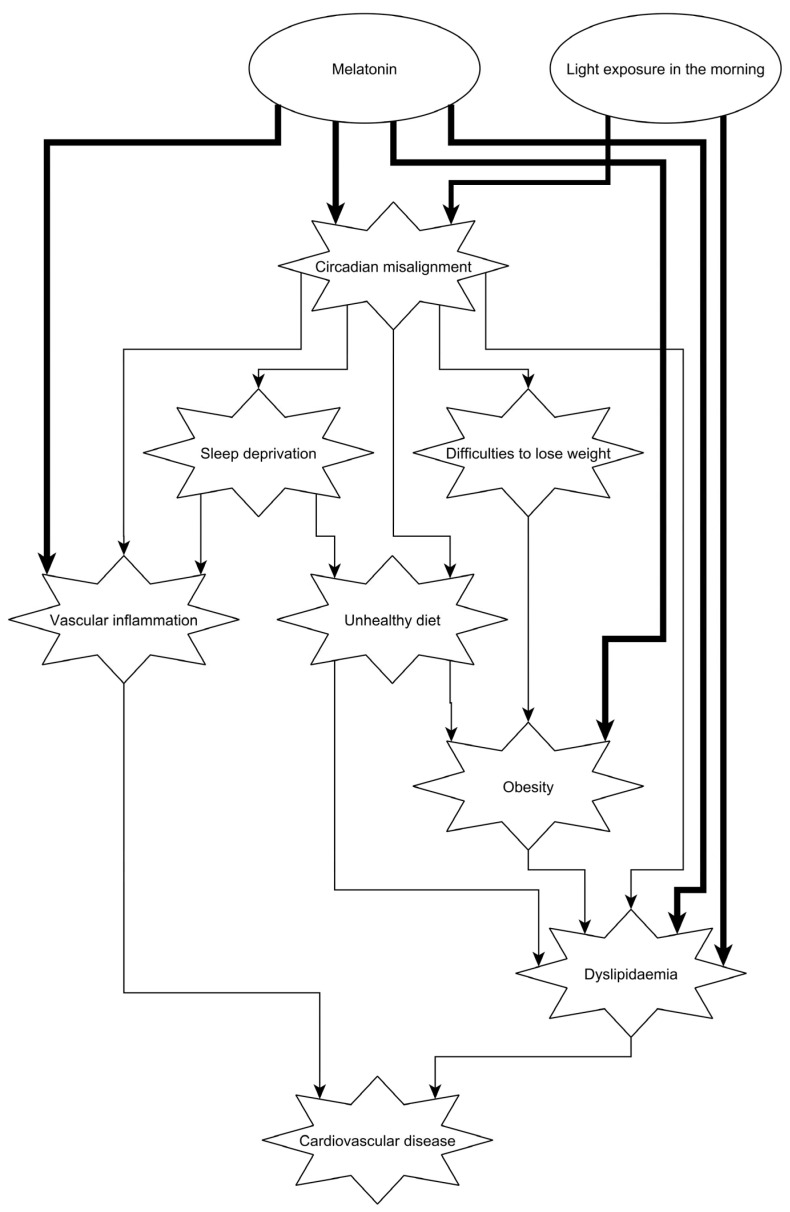
Schematic representation of how circadian misalignment can lead to cardiovascular disease. Circadian misalignment induces dyslipidaemia and vascular inflammation directly, and indirectly through sleep deprivation, diet and obesity (thin lines). Clinical interventions, such as bright light therapy in the morning and melatonin in the evening, may be beneficial to improve the cardiovascular risk (thick lines). For references, please see the text.

## Data Availability

Not applicable.

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
