# Peer review of "The Role of the Circadian Rhythm in Dyslipidaemia and Vascular Inflammation Leading to Atherosclerosis"

_ijms, 2023, doi:10.3390/ijms241814145_

Round 1

Reviewer 1 Report

Reviewer comments and suggestions

The authors in this review highlighted the importance of circadian rhythm on cardiovascular health, focusing on dyslipidaemia and vascular inflammation, two hallmarks of atherosclerosis. Additionally, they discussed the effects of chronotherapy, melatonin, and bright light on dyslipidaemia and vascular inflammation in an effort to propose an agenda for future research and help clinical practice.

Overall, the manuscript needs to be modified. I have listed the concerns/comments that needed to be explained/modified. 

  1. In line 27 “delayed phase sleep disorder,” Could you explain the condition for the common reader of your MS
  2. Comments for Figure 1 From where the figure was adopted, it needs to mention in the legend part
  3. Line 42, the circadian system, needs to be explained well
  4. Lines 53-57 Need proper reference for this line
  5. Line 72 What are ROR
  6. Line 82 References: [4] and provided in the text. 82
  7. Line 83 Before coming to the points the authors need to present a story or paragraph and then start with CVD
  8. Line 102-105 These explanations needs a suitable reference
  9. Section 2.2 The section needs to be explained well,, not sufficient
  10. Line 176-177 So how the authors explain the study
  11. Line 199-202 Did the authors find only one study? Sometimes, metaanalysis is important to discuss these types of work, I saw reference 62 (it's not recent as you mention)
  12. Please avoid big sentences (lines 230-233) and many more
  13. Line 252-254 What would be the levels the authors could explain it
  14.  One table is required in the manuscript to minimize the length of the text
  15. Comments for section 4.1 is this section was important as the 4.2 already shows a similar pattern of text recognized by the section named 4.
  16. Comments for section 5 Do you add this point in the abstract part, please add
  17. In line 503, the authors mention numerous studies but add only one reference to cite
  18. All references need to be modified based on MDPI guidelines.

Author Response

Comment: In line 27 “delayed phase sleep disorder,” Could you explain the condition for the common reader of your MS.

Response: Thank you for the comment. We have rephrased our paragraph.

Comment: Comments for Figure 1 From where the figure was adopted, it needs to mention in the legend part

Response: The corresponding senior author (Andras Bikov), a clinician expert in sleep medicine, drafted the figure himself on a typical sleep wake schedule of a healthy subject and of a patient with DPSD.

Comment: Line 42, the circadian system, needs to be explained well

Response: Thank you. This paragraph has been expanded.

Comment: Lines 53-57 Need proper reference for this line

Response: Thank you. A comprehensive review on circadian rhythm and the zeitgebers by Dijk and Lockley was added as reference.

Comment: Line 72 What are ROR

Response: Sorry for the error, this has now been explained.

Comment: Line 82 References: [4] and provided in the text. 82

Response: We have clarified this in the revise manuscript.

Comment: Line 83 Before coming to the points the authors need to present a story or paragraph and then start with CVD

Response: We have expanded the paragraphs before and after line 83. We believe these changes will improve the readability of the manuscript.

Comment: Line 102-105 These explanations needs a suitable reference

Response: Thank you. We have reorganised this section. The explained mechanisms became part of the text rather than the figure. At this point we wanted to give a brief explanation of atherosclerotic plaque formation and referred the readers on comprehensive review papers. The detailed explanation, how circadian rhythm influences vascular inflammation, is given at later parts and is supported by references.

Comment: Section 2.2 The section needs to be explained well,, not sufficient

Response: We agree that section 2.2. was not as detailed as other sections of the manuscript. To keep the focus on dyslipidaemia and vascular inflammation, we merged this section with 2.1.  

Comment: Line 176-177 So how the authors explain the study

Response: Thank you. BMAL1 controls the transcription of multiple genes which themselves have pleiotropic effect. Therefore, the effect of BMAL1 knock out may not necessarily represent a direct effect of BMAL1 on expression. We believe that the highlighted sentence by the Reviewer was misleading and creates more confusion than clarity, therefore it was deleted. 

Comment: Line 199-202 Did the authors find only one study? Sometimes, metaanalysis is important to discuss these types of work, I saw reference 62 (it's not recent as you mention)

Response: We have extended the manuscript with a recent meta-analysis on evening chronotype and cardiovascular risk.

Comment: Please avoid big sentences (lines 230-233) and many more

Response: Thank you. We have shortened this sentence and other sentences throughout the manuscript.

Comment: Line 252-254 What would be the levels the authors could explain it

Response: The concentration of hormones regulating appetite show diurnal rhythm. The levels of leptin are the highest, the levels of ghrelin are the lowest at night to adjust for metabolic need.

Comment: One table is required in the manuscript to minimize the length of the text

Response: Thank you. We agree that the article was lengthy, therefore a figure summarising the main results was provided (Figure 4). We critically reviewed the paper and excluded the redundant information, making the results more concise. As a result, we believe that a table is no longer needed as the information is better summarised in text.

Comment: Comments for section 4.1 is this section was important as the 4.2 already shows a similar pattern of text recognized by the section named 4.

Response: Section 4 and its subsections were critically reviewed, and the redundant information was deleted.

Comment: Comments for section 5 Do you add this point in the abstract part, please add

Response: Thank you, based on your and other reviewers’ comments, the abstract was expanded including a better summary of section 5.

Comment: In line 503, the authors mention numerous studies but add only one reference to cite

Response: We referred to the comprehensive review by Carrillo-Vico et al. (ref 202) in the next sentence. The diverse role is further explained in the forthcoming paragraphs citing individual studies.

Comment: All references need to be modified based on MDPI guidelines.

Response: The reference style template provided on the journal homepage was used to create reference list.

Reviewer 2 Report

"The role of the circadian rhythm in dyslipidaemia and vascular inflammation leading to atherosclerosis" is an interesting and important paper and should be published after minor revision.

The abstract should be extended, conclusion is missing and linguistic style should be improved.

"The role of the circadian rhythm in dyslipidaemia and vascular inflammation leading to atherosclerosis" is an interesting and important paper and should be published after minor revision, esp.

the linguistic style should be improved.

Author Response

Comment: The abstract should be extended, conclusion is missing and linguistic style should be improved.

Response: Thank you. Based on your comment the abstract was extended. Please, find conclusions in section 6 (we have renamed it to better serve the purpose. The linguistic style was critically reviewed and we have made corrections throughout the manuscript.

Reviewer 3 Report

Dear Editor and Authors,

With great interest I have read the manuscript " The role of the circadian rhythm in dyslipidemia and vascular inflammation leading to atherosclerosis"

This review manuscript is well organized and comprehensively described.This work represents a significant contribution to the field and reveals novel findings in this topic.

However, important considerations should be changed before it can be further processed:

- Figure explanation should be re-written in shorter way, not as another whole paragraph (discussion and explanation should be written in other parts of the manuscript)

- Since the aim and the title of the journal, this review does not fit completely, it is more clinical review with several molecular explanations. 

English is acceptable overall

Author Response

Comment: - Figure explanation should be re-written in shorter way, not as another whole paragraph (discussion and explanation should be written in other parts of the manuscript)

Response: Thank you. Following your advice, the explanations of figures became part of the text rather than the figures 1 and 3.  

Comment: - Since the aim and the title of the journal, this review does not fit completely, it is more clinical review with several molecular explanations.

Response: Thank you very much for your comment. The main aim of the article was to bring clinicians attention to molecular mechanisms involved in circadian misalignment-induced cardiovascular disease.